# Systolic Pulmonary Artery Pressure as Long-Term Mortality Predictor in Elderly Critically Ill with Severe COVID-19 Pneumonia

**DOI:** 10.3390/v17020244

**Published:** 2025-02-11

**Authors:** Marko Kurnik, Helena Božič, Matej Podbregar

**Affiliations:** 1Department of Internal Intensive Medicine, General Hospital Celje, 3000 Celje, Slovenia; helena.bozic@sb-celje.si (H.B.); matej.podbregar@sb-celje.si (M.P.); 2Faculty of Medicine, University of Ljubljana, 1000 Ljubljana, Slovenia

**Keywords:** COVID-19, elderly, ICU, long-term mortality, echocardiography, systolic pulmonary artery pressure, sPAP, PAPS, pulmonary hypertension

## Abstract

Background: COVID-19 can cause acute pulmonary hypertension (PH), worsening outcomes in critically ill elderly patients. Point-of-care ultrasound (POCUS), assessing right ventricular hemodynamics, predicts short-term outcomes. This study examines the long-term impact of acute PH on mortality in elderly COVID-19 patients. Methods: This retrospective long-term study analyzed data from patients over 70 years old with severe COVID-19 pneumonia admitted to a mixed 25-bed, level 3 intensive care unit (ICU). POCUS focused on systolic pulmonary artery pressure (sPAP) at admission. Mortality was evaluated 1000 days post-admission. Results: The study included 130 patients, comprising 30 long-term survivors and 100 non-survivors, with a cumulative long-term mortality rate of 77%. Non-survivors had significantly higher sPAP values (39.1 ± 12.8 vs. 30.4 ± 9.2, *p* = 0.04), which were associated with long-term mortality in survival analysis. Conclusion: Acute pulmonary hypertension (PH), reflected by elevated systolic pulmonary artery pressure (sPAP), is strongly associated with long-term mortality in elderly critically ill COVID-19 patients. Early assessment of sPAP via POCUS may help identify high-risk patients and guide management strategies to improve outcomes.

## 1. Introduction

Coronavirus disease 2019 (COVID-19) is a multi-system disease caused by the severe acute respiratory syndrome coronavirus 2 (SARS-CoV-2) [1].

Pulmonary hypertension (PH) is defined by a rise in arterial pressure of the pulmonary vasculature which represents right ventricular afterload pressure. PH is divided into five groups that determine the therapeutic approach and is strictly determined by obtaining invasive hemodynamic measurements [2]. Invasive measurements are especially important in the evaluation of chronic PH which is defined by mean pulmonary artery pressure (mPAP) > 20 mmHg. Before performing invasive procedures, clinical suspicion of PH is based on a history of progressively worsening dyspnea on exertion, with or without syncope, enlarged liver/spleen, peripheral edema, and cyanosis. The screening method of choice is echocardiography. By measuring peak tricuspid regurgitation velocity (pTRV), the estimated systolic pulmonary arterial pressure (sPAP) is calculated, and PH is deemed unlikely (pTRV ≤ 2.8 m/s, sPAP ≤ 32 mmHg), likely (pTRV 2.9–3.4 m/s, sPAP 33–46 mmHg), or very likely (pTRV ≥ 3.4 m/s, sPAP ≥ 46 mmHg). Echocardiography is even more important in determining the presence of acute PH, with acute respiratory distress syndrome being the third most frequent cause (after pulmonary embolism and sepsis) [3,4]. In critically ill COVID-19 patients, PH manifests as an acute presentation in previously healthy individuals or as an acute exacerbation of pre-existent chronic PH and is predictive of worse short-term outcomes, especially in the elderly [5]. New onset chronic PH after COVID-19 pneumonia has been reported [6].

During the pandemic, in-hospital mortality among elderly, especially frail, COVID-19 patients was high, with the main factors contributing to worse outcomes being diabetes, chronic kidney disease, arterial hypertension, and obesity [7]. Bedside echocardiography was a pivotal clinical tool in evaluating acute cardiac pathology with a focus on hemodynamics [8].

Acute COVID-19-associated pulmonary hypertension, measured noninvasively, has been shown to be a predictor of short-term mortality [9]. The aim of the current study is to evaluate the effect of acute pulmonary hypertension on long-term mortality in elderly critically ill COVID-19 patients.

## 2. Materials and Methods

### 2.1. Setting

A retrospective cohort study was conducted in a mixed 25-bed, level 3 intensive care unit (ICU) at the General and Teaching Hospital Celje, Slovenia, on a population of patients admitted during a 6-month period (from October 2020 to April 2021), with long-term mortality data collected one thousand days after the admission date for each patient.

### 2.2. Patients

The ICU was designated exclusively for the treatment of adult (≥18 years old) critically ill patients with proven SARS-CoV-2 infection. The diagnosis of COVID-19 was established by at least one positive real-time polymerase chain reaction (RT-PCR) test for SARS-CoV-2 on one of the respiratory specimens (nasopharyngeal swab, sputum, and/or lower respiratory tract specimens).

Only elderly (≥70 years old) critically ill patients were included in the final analysis. Patients with a known history of systolic left heart failure (severely diminished left ventricular ejection fraction, <30%), right heart failure, and/or previously detected pulmonary hypertension (estimated sPAP >35 mmHg) were excluded from the study. Pregnant women were not treated in our ICU.

Data were collected after obtaining approval from the Republic of Slovenia National Medical Ethics Committee (No. 0120-168/2021/7, 22 July 2021) and from the Institutional Review Board of General Hospital Celje (No. 17/KS/2021-1, 5 March 2021). Informed consent was waived due to the retrospective nature of the study.

### 2.3. Patient Data

The following patient data were collected from the digital hospital information system BIRPIS21 (SRC Infonet, Kranj, Slovenia): basic demographic data, previous medical history, and chronic illnesses (e.g., malignant diseases, arterial hypertension, diabetes, heart failure, chronic obstructive pulmonary disease, and chronic renal disease).

### 2.4. Laboratory

The General Laboratory of our institution conducted the majority of the laboratory analyses. The focus was on admission laboratory values and the worst laboratory values recorded during the entire ICU stay (e.g., lowest pH, highest pCO_2_, lowest pO_2_, highest D-dimer, highest troponin T, highest procalcitonin (PCT), highest C-reactive protein (CRP), highest creatinine, highest leukocyte count, etc.).

### 2.5. Echocardiography

A transthoracic echocardiographic examination was conducted using a cardiac probe on a GE Vivid S60 Ultrasound machine (GE Healthcare, Chicago, IL, USA) by an intensive care specialist upon patient admission. Admission echocardiography data, recorded according to protocol, were documented in the Centricity Critical Care information system (GE Healthcare, USA). The examination was also recorded for subsequent offline analysis using the GE EchoPAC Clinical Workstation Software v.204 rev.41.4 (GE Healthcare, USA). This analysis was performed by an experienced ICU specialist with expertise in cardiology (MP). Collected data included left ventricular ejection fraction (LVEF) assessed by eyeballing, velocity time integral (VTI) in the left ventricular outflow tract (LVOT), tricuspid annular plane systolic excursion (TAPSE), minimum and maximum inferior vena cava diameters (VCI min and VCI max, respectively), and systolic pulmonary artery pressure (sPAP), which was calculated using the Bernoulli equation, based on the maximum velocity of tricuspid regurgitation and invasively measured central venous pressure (CVP). Morphology measurements were not precisely obtained because the examinations were conducted with patients in supine or partial left lateral recumbent positions, which are suboptimal. Additionally, invasive mPAP measurements using a pulmonary artery catheter (Swan-Ganz catheter) were not performed. Definitions of right ventricular dysfunction followed the American Heart Association Guidelines [10]. An sPAP threshold of 35 mmHg for diagnosing pulmonary hypertension was established based on a prior study examining 1-year mortality in patients with chronic lung disease [11]. This threshold is further supported by Doppler measurements in healthy individuals, which identified a tricuspid gradient of 30 mmHg as the upper limit of normal [12].

### 2.6. Treatment

Treatment data were extracted from the intensive care information system (Centricity Critical Care, GE Healthcare, USA). All patients initially received methylprednisolone at a dose of 1 mg/kg of body weight per day, following the protocol in use at the time [13].

To assess respiratory support, the following information was gathered: use of self-proning and proning during mechanical ventilation; frequency and duration of high-flow nasal cannula (HFNC) oxygen therapy, non-invasive ventilation (NIV), and invasive mechanical ventilation (IMV). The duration of HFNC and NIV was guided by the ROX index [14]. For patients requiring IMV, data on the maximum positive end-expiratory pressure (PEEP), maximum peak pressure, and highest tidal volume recorded during treatment were documented. Additionally, the use of nitric oxide inhalation therapy, norepinephrine, levosimendan, systemic thrombolytic therapy, and renal replacement therapy was recorded.

### 2.7. Complications and Mortality

Data on complications (e.g., ventilator-associated pneumonia (VAP), catheter-related bloodstream infections, urosepsis, and fungal infections), as well as ICU and hospital mortality, were collected from the intensive care and hospital information systems.

Long-term mortality data were obtained from the national health insurance database upon individual patient request.

### 2.8. Definitions

Definitions of VAP, catheter-related bloodstream infection (CRBSI), and fungal infection remain as previously described [5].

### 2.9. Primary Outcome

The primary outcome assessed was all-cause long-term mortality.

### 2.10. Secondary Outcome

The secondary outcome assessed was mortality at ICU discharge.

### 2.11. Sample Size Estimation

To achieve 80% power with a type I error rate of 0.05 (two-tailed) for detecting statistically significant differences in PASP between long-term survivors and non-survivors, a sample size of 14 patients in the survivor group and 42 patients in the non-survivor group was estimated, based on a 1:3 ratio between the groups. This assumption was extrapolated from the findings of Minkin et al. [15], who reported mortality rates of 80% and 43.8% for adult patients with and without pulmonary hypertension concomitant with COVID-19 infection. The study group size was doubled due to the high ICU mortality rate in our population.

### 2.12. Statistical Analysis

Data were summarized as mean (±standard deviation) for continuous variables and as absolute and relative frequencies for categorical variables. Tests for normality were conducted on continuous variables. Student’s *t*-test was used for continuous variables, and the Chi-square test was used for categorical data. Univariate and multivariate logistic regression modeling, with odds ratio calculations, was employed to test the relationship between echocardiography, length of stay (LOS) data, complications, and ICU/hospital mortality. The analyses were performed using SPSS v.25.0 (SPSS Inc., Chicago, IL, USA) and MedCalc v.19.1.3 (MedCalc Software Ltd., Ostend, Belgium). A *p*-value of < 0.05 was considered statistically significant. Cox proportional hazards regression and Kaplan-Meier survival analysis were used for long-term mortality assessment.

## 3. Results

During the study period, 371 patients were admitted to the ICU. Of these, 221 were excluded as they were younger than 70 years, and 29 were excluded based on a history of pulmonary hypertension and/or diminished left ventricular systolic function. One patient was admitted due to iatrogenic pneumothorax during pacemaker implantation and was subsequently excluded from the analysis because only a positive SARS-CoV-2 nasopharyngeal swab was detected, with no evidence of COVID-19 pneumonia. There was an overlap of 10 patients who were younger than 70 years and had existing comorbidities as defined in the methods. A total of 130 patients were included in the final analysis, consisting of 30 long-term survivors (at 1000 days after ICU admission) and 100 long-term non-survivors. The long-term mortality rate was 77%. A general description of patients, previous history, and chronic therapy is presented in Table 1. Tests for normality did not reject the null hypothesis that variables were normally distributed. There was a significant difference in average age between survivors and non-survivors (74 ± 4 vs. 76 ± 5 years, *p* < 0.01). Length of ICU treatment was not significantly different between groups (8.7 ± 7.5 vs. 11.3 ± 12.0 days, *p* = 0.26) (Table 1).

Vital parameters measured at admission were not significantly different between groups. The PaO_2_/FiO_2_ ratio at admission was not predictive of long-term outcome (63.9 ± 43.0 vs. 74.2 ± 58.2, *p* = 0.37) (Table 2).

Lower values of pH, higher pCO_2_, lower hemoglobin oxygen saturation (StHbO_2_), higher leukocyte count, higher platelet count, higher creatinine levels, and higher D-dimer recorded during the ICU stay were predictive of worse long-term survival (Table 2). Higher procalcitonin (PCT) levels were predictive of worse long-term outcomes, in contrast to higher C-reactive protein (CRP), which was not (Table 2).

Per protocol, all patients were treated with methylprednisolone (see Methods, Treatment subsection) (Table 3).

Neither the time of invasive mechanical ventilation (IMV) nor its frequency were predictive of long-term outcomes, nor were the time of non-invasive mechanical ventilation or the frequency/time of high-flow nasal cannula (HFNC) oxygenation. Long-term non-survivors were more frequently supported with NIV than survivors (37 [37%] vs. 4 [13%], *p* = 0.01). There was no difference between groups in maximal PEEP, maximal peak pressure, or tidal volume recorded during the ICU stay. Complication rates were not associated with long-term survival (Table 3). A comparison of VAP incidence in the groups of survivors and non-survivors who were mechanically ventilated showed no difference (38% vs. 54%, *p* = 0.28).

A bedside echocardiogram was performed upon admission to the ICU (Table 4). There were no differences between groups in LVEF or systolic VTI measured in the LVOT. sPAP at admission was lower in long-term survivors compared to non-survivors (30.4 ± 9.2 vs. 39.1 ± 12.8 mmHg, *p* = 0.04). Additionally, TAPSE was significantly higher in long-term survivors (22.5 ± 3.5 vs. 18.7 ± 5.1 mm, *p* < 0.01). Lung ultrasound findings did not show significant differences between groups. At admission, pulmonary embolism was confirmed via computed tomography pulmonary angiography (CTPA) in 13 patients, including 3 long-term survivors (10%) and 10 long-term non-survivors (10%) (*p* = 1.00). All cases of pulmonary embolism were confined to segmental (affecting a maximum of two segmental arteries) or subsegmental arteries.

Since the population was expanded from our original report [5] analyzing the relationship between sPAP and ICU mortality, we confirm that the significance of the analyzed variables did not change for ICU mortality between populations. sPAP remained a statistically significant predictor of ICU mortality in both univariate and multivariate logistic regression models (OR 1.058, 95% CI 1.011–1.108, *p* = 0.016 and OR 1.057, 95% CI 1.005–1.112, *p* = 0.032 with model Chi-squared *p* < 0.008). The sPAP cut-off value, determined using receiver operating characteristic (ROC) curves, dropped from the originally reported >42 mmHg of estimated systolic pulmonary artery pressure to >38 mmHg (area under the ROC curve 0.676, 95% CI 0.549–0.787, *p* = 0.008).

Regarding the relationship between sPAP at ICU admission and long-term survival, an sPAP of up to 35 mmHg is associated with reduced mortality (Figure 1). More than 90% of mortality events occurred within the first 200 days following ICU admission.

When modeling long-term survival with age as an additional variable (which differed significantly between groups), sPAP remained significant, while age was not significant (Figure 1 and Table 5).

The results of ROC curve analysis for sPAP as a long-term mortality predictor are presented in Table 6.

## 4. Discussion

Our follow-up study confirmed that sPAP at ICU admission is not only a predictor of short-term mortality but also of long-term mortality in elderly critically ill patients with COVID-19 pneumonia.

Standard protocol at our ICU during the COVID-19 pandemic mandated bedside echocardiography examinations for all admitted patients. As a result, robust and easily obtainable parameters of left ventricular (VTI LVOT, estimated LV EF) and right ventricular (TAPSE, sPAP, relative size to LV) were obtained. The focus of right ventricular hemodynamics was in accordance with well-known pulmonary vasculature dysfunction in general ARDS patients [16]. Advanced parameters of cardiac morphology and function were not routinely obtained (e.g., three-dimensional echocardiography, speckle tracking) as ideal projections are difficult to visualize at the bedside in critically ill. In our analysis, the focus was on recorded data that could be obtained in most modern ICU units [17].

Multiple factors contribute to elevated pulmonary pressure. Pulmonary hypertension (PH) due to left heart disease falls under class 2 of the classification proposed by the World Symposium on Pulmonary Hypertension Association [2]. In our study, we specifically selected patients without advanced systolic left ventricular dysfunction. While advanced diastolic dysfunction was not explicitly defined as an exclusion criterion, it was absent in the known history of all patients. However, the assessment of diastolic function at admission was not included in our protocol. Pulmonary embolism (PE) and hypoxia are more relevant to our study. At admission, the PaO_2_/FiO_2_ ratio in our cohort was consistent with severe ARDS in both groups, with no significant difference between them. Acute hypoxia can induce rapid vasoconstriction of the pulmonary vascular bed, which reverses swiftly and completely once oxygen levels are restored [18]. Acute PE is another factor that can cause a sudden rise in pulmonary artery pressure, ranging from subclinical effects to acute hemodynamic deterioration. Significant elevation of pulmonary artery pressure typically occurs only when angiographic obstruction of the pulmonary arteries exceeds 50% [19,20]. In our study, 13 patients had small pEs, and it is highly unlikely that hypoxia or the presence of PE significantly influenced the differences in sPAP between the groups. No significant differences in the PaO_2_/FiO_2_ ratio or the incidence of PE were observed between the groups, indicating that these factors were uniformly distributed. Moreover, all cases of pulmonary embolism observed were classified as small, involving less than 50% of the vascular bed.

### 4.1. Short-Term Versus Long-Term Study

Regarding differences with our previous study on sPAP as a short-term ICU mortality predictor [5], there are some important differences.

Long-term non-survivors were older than long-term survivors, even in the currently selected elderly population (77.6 ± 5.0 vs. 74.2 ± 3.0 years, *p* < 0.01), in contrast to age not being significantly different between groups for short-term survival. Age was not associated with short-term mortality in a small study of elderly patients from long-term care facilities in Italy [21], but was associated with long-term mortality in a non-selected group of patients, as shown in larger prospective [22] and retrospective studies [23].

ICU LOS was not associated with long-term mortality, in contrast to short-term mortality. Interestingly, while chronic obstructive pulmonary disease (COPD) and chronic kidney insufficiency (CKI) were not associated with ICU mortality, they were both associated with long-term mortality. The finding for CKI was also confirmed in the aforementioned study by Santos et al., but the same is not true for COPD. The population was larger, although the long-term follow-up was shorter. Neither was associated with short-term mortality [22]. PaO_2_/FiO_2_ ratio (P/F ratio) at admission is not predictive of either short or long-term mortality. Most patients admitted to our department were receiving 100% oxygen through a non-rebreather mask. All patients had already developed severe acute respiratory distress syndrome (ARDS) according to the Berlin definition [24], which may explain the uniformity between groups.

The highest C-reactive protein (CRP) value was significantly associated with short-term mortality (*p* < 0.001), while procalcitonin (PCT) showed no such association (*p* = 0.3). In contrast, the pattern reversed for long-term mortality: CRP was not significantly associated (*p* = 0.07), whereas PCT demonstrated a significant association (*p* = 0.02). Procalcitonin has been identified as a strong predictor of ICU admission and/or mortality in numerous studies [25,26,27]. A study by colleagues from the large multicentric international SepsEast Registry [25] showed differences in PCT levels on admission between ICU survivors and non-survivors (0.30 vs. 0.41, respectively, *p* < 0.01), even though both values were below the prognostic threshold of sepsis [28]. In our case, PCT seems to indicate the presence of sepsis or bacterial superinfection, as it is known that sepsis is an independent predictor of long-term mortality [29]. However, it must be noted that the incidence of microbiologically proven complications did not differ between groups in our current long-term analysis.

Invasive mechanical ventilation (IMV) is not a predictor of long-term mortality, as it was for short-term mortality (*p* = 0.07 and *p* < 0.01, respectively). A significant difference is observed in non-invasive ventilation between long-term survivors and non-survivors (13% vs. 37%, *p* = 0.01). This finding aligns with analyses showing the deleterious effects of delayed intubation in large propensity-matched studies, where patients with delayed intubation had significantly higher mortality rates across multiple timeframes: hospital mortality (27.3% vs. 37.1%, *p* = 0.01), ICU mortality (25.7% vs. 36.1%, *p* = 0.007), and 90-day mortality (30.9% vs. 40.2%, *p* = 0.02) [30]. Similarly, smaller retrospective studies have shown that the hazard ratio (HR) for mortality with invasive ventilation after ≥120 h from hospital admission compared to <120 h was 2.03 (*p* = 0.02) [31]. There were no differences in mechanical ventilation parameters between the groups.

Regarding the systolic function of the left ventricle at admission, the groups did not differ significantly. All patients with pre-existing and acute left ventricular failure (LVEF <30%) were excluded per the study design. Systolic function was preserved in both groups, indicating the absence of acute cardiac depression, which can be observed in septic shock patients [32].

Lung ultrasound (LUS) examinations did not show significant differences in the long-term analysis. This contrasts with our short-term study, which found an association between a diffuse B-lines pattern and increased mortality. While a LUS score has been identified as a predictor of COVID-19 mortality in a prospective study in the emergency department [33], long-term data remains scarce. Our study did not find any association between lung ultrasound findings and long-term mortality in critically ill elderly COVID-19 patients.

### 4.2. Systolic Pulmonary Artery Pressure and Long-Term Mortality

Pulmonary vascular dysfunction in ARDS, which leads to acute pulmonary hypertension and right ventricular failure (acute cor pulmonale), is a well-recognized phenomenon with a known negative impact on mortality [34,35,36]. Higher values of invasively measured mean systolic pulmonary pressure have been associated with 90-day mortality in a study of 145 ARDS patients (age 61 ± 15 years) [37]. A recent large population study (156,687 patients) analyzing hospitalization outcomes found that patients with ARDS and coexisting pulmonary hypertension (PH) had higher in-hospital mortality (OR 1.52, 95% CI 1.46–1.58, *p* < 0.001) [38]. On the other hand, a pre-COVID-19 meta-analysis of nine studies found that right ventricular (RV) injury (defined as RV dysfunction, RV dysfunction with hemodynamic compromise, RV failure, or acute cor pulmonale) was associated with overall and short-term mortality (OR 1.45, 95% CI 1.13–1.86 and OR 1.48, 95% CI 1.14–1.93), but no such association was observed for long-term mortality (OR 1.24, 95% CI 0.66–2.33), defined as mortality beyond 30 days [39]. In studies unrelated to COVID-19 and ARDS, Barywani reported that a systolic pulmonary artery pressure (sPAP) >35 mmHg was associated with higher 5-year mortality in octogenarian patients with congestive heart failure (CHF) or acute coronary syndrome (ACS) (HR 1.7, 95% CI 1.1–2.6, *p* = 0.013) [40]. Additionally, the TAPSE/sPAP ratio has been found to be independently associated with one-year all-cause mortality in septic shock patients (HR 0.007, 95% CI 0.000–0.162, *p* = 0.002) [41]. Despite this, studies on the long-term effects of acute pulmonary hypertension in ARDS patients on mortality remain scarce.

A similar effect of pulmonary hypertension (PH) has been observed in ARDS following severe SARS-CoV-2 infection. Researchers from New York, USA, confirmed the association between PH in COVID-19 pneumonia and worse short-term mortality in a diverse, unselected patient population. They found no differences in ICU admission, presence of ARDS, ICU length of stay (LOS), mechanical ventilation (MV), or hospital LOS between groups, but a significant increase in mortality in the PH group compared to the non-PH group (80% vs. 43.8%, *p* < 0.01). Notably, patients who underwent echocardiograms had higher mortality than those who did not (59.7% vs. 32.3%, *p* < 0.001), suggesting that focused care is often directed toward patients with more severe disease [15]. Papageorgiou et al. conducted a comprehensive meta-analysis involving 3373 patients, examining PH and its association with mortality and ICU admission in SARS-CoV-2 patients. They found significantly higher odds for all-cause mortality in patients with PH compared to those without PH (OR 3.89, 95% CI 2.85–5.31, *p* < 0.001) [9]. This finding aligns with our short-term study [5] where systolic pulmonary artery pressure (sPAP) was a predictor of ICU and hospital mortality in both univariate (OR 1.056, 95% CI 1.005–1.111 and OR 1.068, 95% CI 1.007–1.133, respectively) and multivariate models (OR 1.061, 95% CI 1.003–1.124 and OR 1.073, 95% CI 1.003–1.146, respectively). While numerous studies have been conducted on right ventricular function and pulmonary vasculature resistance in relation to short-term mortality, long-term follow-up data in COVID-19 patients remains scarce.

Systolic pulmonary artery pressure (sPAP) at ICU admission is significantly associated with long-term mortality in our current study of elderly critically ill COVID-19 patients. The overall mortality in our cohort is high. At 1000 days, 91% of patients in the pulmonary hypertension (PH) group (sPAP at ICU admission ≥ 35 mmHg) and 78% of patients in the non-PH group (sPAP at ICU admission < 35 mmHg) had died. In the Cox proportional-hazards regression survival analysis, sPAP greater than 35 mmHg is associated with worse long-term outcomes. When age is included as a covariate, sPAP remains statistically significant, while age does not (Figure 1 and Table 5). The authors of a long-term study reported higher mortality in an elderly subgroup (defined as those aged ≥ 65 years) compared to a non-elderly group (46.06% vs. 11.65%, *p* < 0.01, respectively). Although the mortality in our cohort is significantly higher, this can be attributed to our focus on critically ill elderly patients. Additionally, the study period in our cohort is longer, though more than 90% of all events occurred within the first 200 days. Another study found NT-proBNP and troponin levels to be statistically significant predictors of 1-year mortality in a multivariate analysis, although no right ventricular hemodynamic parameters were reported [42].

No long-term studies have explored the relationship between TAPSE or sPAP and COVID-19 mortality.

### 4.3. Long-Term Follow-Up of the Survivors

In addition to the statistical analysis of data collected during the ICU stay, a follow-up of long-term survivors was conducted. This involved reviewing electronic medical records from our hospital and directly contacting patients for additional information. Of the 30 long-term survivors, data could not be obtained for 5 patients, and 1 patient had subsequently died from aggressive hepatic cancer. Among the survivors, 25% belonged to the elevated sPAP group (defined as sPAP ≥ 35 mmHg). While the data collected are not standardized enough for objective statistical analysis, a narrative review can be presented.

Five patients, all from the non-elevated sPAP group, developed paroxysmal atrial fibrillation requiring lifelong anticoagulation therapy. Fifteen patients were followed by outpatient pulmonologists, of whom nine experienced complete regression of opacities on chest X-ray—all from the non-elevated sPAP group. Six patients had residual interstitial changes on chest X-ray; among these, four were confirmed to have interstitial fibrosis on high-resolution computed tomography (HRCT). Of those with interstitial fibrosis, two required long-term supplemental oxygen for 1–2 years, and three (including both requiring oxygen supplementation) were from the elevated sPAP group. Three patients developed secondary adrenal insufficiency following high doses of methylprednisolone. Two recovered within 1.5 years, while one continued to show an abnormal adrenocorticotropic hormone stimulation test almost four years later. Additionally, two patients developed symptoms of depression or anxiety necessitating pharmacological intervention, both of whom had required mechanical ventilation. One patient developed ocular myasthenia gravis six months after ICU discharge, which later progressed to the generalized form. This required intensive care treatment, multiple plasmapheresis sessions, and long-term treatment with efgartigimod alfa.

Twelve patients underwent outpatient follow-up echocardiography, including three from the elevated sPAP group. Among these, one patient showed signs of elevated pulmonary pressure three months after hospital discharge, which completely resolved by the six-month follow-up. The other two patients in the elevated sPAP group had no signs of pulmonary hypertension on follow-up, one at six months and the other at two years post-discharge. None of the patients in the non-elevated sPAP group exhibited signs of pulmonary hypertension during follow-up echocardiography. However, one patient underwent elective percutaneous coronary intervention nine months post-discharge due to persistent exertional dyspnea despite normal cardiac ultrasound results. Another patient was diagnosed with three-vessel disease more than three years after hospital discharge, following the development of exertional angina symptoms and a decline in left ventricular ejection fraction (from normal at six months to moderately impaired). This patient also experienced progression of diastolic dysfunction from grade I to grade III over three years, although no signs of pulmonary hypertension were observed.

Our long-term follow-up findings align with studies on post-COVID depression [43], atrial arrhythmias [44], and radiographic changes [45]. While some research has been conducted on long-term cardiac sequelae [46], the full extent of the consequences of the COVID-19 pandemic remains to be uncovered.

### 4.4. Study Limitations

The first limitation of our study is its retrospective nature, which only allowed for the analysis of routinely obtained data. Although echocardiographic data was reevaluated (M.P.), the positioning of patients was not standardized, which could introduce slight irregularities in the obtained projections. However, this limitation is balanced by the fact that these image/data acquisition methods can be implemented in any modern ICU. Secondly, we recognize that the use of more advanced echocardiographic techniques could potentially reveal subtler myocardial changes, but such techniques are challenging to deploy or obtain in an intensive care setting. Comparison to invasive pulmonary artery measurements would also be extremely valuable, but pulmonary artery catheterization was not routinely performed in our department during the COVID-19 pandemic. A third limitation is the relatively low number of included patients, which limited our ability to construct and test more complex models or include a broader age range. Finally, the single-center design of the study is another limitation, as focusing on a single department may not fully account for potential internal biases or weaknesses.

## 5. Conclusions

Mortality in general ARDS patients remains high, with many experiencing long-lasting sequelae [47]. Numerous studies have been conducted to identify predictors of mortality in COVID-19 patients, particularly regarding right ventricular hemodynamics. However, most of these studies have focused on short-term mortality, as we have previously explored in detail [5].

Our current study demonstrates the continued deleterious effect of elevated sPAP at ICU admission on elderly critically ill COVID-19 patients. We believe that large long-term studies are urgently needed, as a significant portion of the population has been infected with SARS-CoV-2, even though the COVID-19 pandemic is (fortunately) over. While our study focused on extreme cases (only critically ill patients), better long-term data will help healthcare and long-term care providers better direct their efforts.

## Figures and Tables

**Figure 1 viruses-17-00244-f001:**
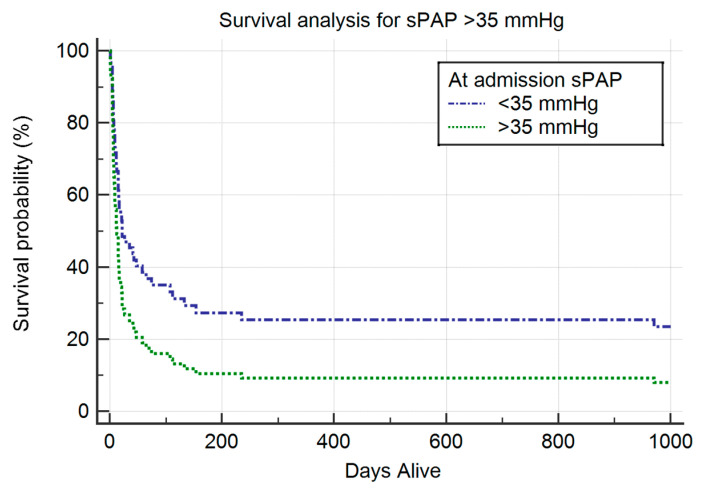
Cox proportional-hazards regression survival analysis.

**Table 1 viruses-17-00244-t001:** General description of patients, previous history, and chronic therapy.

Variable	All (*n* = 130)	Long-Term Survivors (*n* = 30)	Long-Term Non-Survivors (*n* = 100)	Statistics (*p*)
Age, years	76.8 ± 5.0	74.2 ± 3.6	77.6 ± 5.0	<0.01 *
Gender, female/male, *n*	33/97	6/24	27/73	0.49
Height, cm	172.8 ± 7.3	174.1 ± 6.6	172.4 ± 7.5	0.44
Body weight, kg	86.8 ± 15.9	92.7 ± 15.6	85.0 ± 15.6	0.02 *
COVID-19 symptoms ^1^, days	7.5 ± 4.2	8.5 ± 3.9	7.2 ± 4.2	0.15
ICU LOS, days	10.7 ± 11.1	8.7 ± 7.5	11.3 ± 12.0	0.26
Hospital LOS, days	19.7 ± 10.9	22.6 ± 8.2	18.8 ± 11.4	0.10
Previous history				
LV hypertrophy, *n* (%)	27 (21)	8 (27)	8 (19)	0.44
Malignant disease, *n* (%)	25 (19)	5 (17)	20 (20)	0.80
Arterial hypertension, *n* (%)	93 (72)	20 (67)	73 (73)	0.50
Diabetes, *n* (%)	47 (36)	10 (33)	37 (37)	0.83
COPD, *n* (%)	13 (10)	0 (0)	13 (13)	0.04 *
Asthma, *n* (%)	8 (6)	3 (10)	5 (5)	0.39
Chronic kidney failure, *n* (%)	27 (21)	2 (7)	25 (25)	0.04 *
Therapy at home				
Statins, *n* (%)	45 (35)	12 (40)	33 (33)	0.52
Beta-blocker, *n* (%)	50 (39)	8 (27)	42 (42)	0.14
Inhalation corticosteroids, *n* (%)	17 (13)	3 (10)	14 (14)	0.76
ACE inhibitors, *n* (%)	54 (42)	12 (40)	42 (42)	1.00
Insulin, *n* (%)	16 (12)	3 (10)	13 (13)	1.00
Aspirin, *n* (%)	40 (31)	7 (23)	33 (33)	0.37

Values represent means with standard deviations or number of subjects with percentages. * Denotes statistically significant difference between groups at <0.05 level. ^1^ Days of COVID-19 symptoms before ICU admission. Abbreviations: ACE, angiotensin-converting enzyme; COPD, chronic obstructive pulmonary disease; COVID-19, Coronavirus disease 2019; ICU, intensive care unit; LOS, length of stay; LV, left ventricle.

**Table 2 viruses-17-00244-t002:** Clinical parameters on admission and laboratory findings during treatment.

Variable	All (*n* = 130)	Long-Term Survivors (*n* = 30)	Long-Term Non-Survivors (*n* = 100)	Statistics (*p*)
At admission				
Heart rate, bpm	89.7 ± 23.9	85.7 ± 23.7	91.1 ± 23.9	0.29
Respiratory rate, rpm	29.9 ± 7.9	27.3 ± 6.7	30.8 ± 8.1	0.10
SAP, mmHg	140.1 ± 29.0	148.2 ± 27.5	137.4 ± 29.1	0.08
DAP, mmHg	67.2 ± 15.0	69.9 ± 15.3	66.3 ± 14.8	0.24
PaO_2_/FiO_2_, mmHg	71.8 ± 55.0	63.9 ± 43.0	74.2 ± 58.2	0.37
Lactate, mmol/L	2.9 ± 3.1	1.7 ± 0.6	3.3 ± 3.5	0.01 *
Extreme values in ICU				
Highest FiO_2_, %	93 ± 18	91 ± 21	95 ± 14	0.02 *
Lowest pH, value	7.26 ± 0.14	7.34 ± 0.09	7.18 ± 0.13	<0.01 *
Lowest PaO_2_, kPa ^1^	8.02 ± 3.43	8.29 ± 1.90	7.74 ± 4.44	0.40
Highest PaCO_2_, kPa ^1^	8.43 ± 3.48	6.70 ± 2.43	10.09 ± 3.54	<0.01 *
Highest HCO_3_, mmol/L	28.9 ± 6.7	28.9 ± 6.07	30.2 ± 7.13	0.30
Lowest StHbO_2_, %	84.4 ± 8.4	86.9 ± 5.8	78.9 ± 8.6	<0.01 *
Highest creatinine, µmol/L	228.3 ± 195.6	98.6 ±34.7	254.5 ± 209.5	<0.01 *
Highest proBNP, pg/mL ^†^	12680.1 ± 12725.8 (*n* = 21)	4344.0 ± 4665.5 (*n* = 2)	13557.6 ± 13046.0 (*n* = 19)	0.34
Highest troponin I, ng/mL ^†^	409.1 ± 143.7 (*n* = 97)	62.0 ±112.9 (*n* = 20)	499.2 ± 1599.0 (*n* = 77)	0.23
Highest D-dimer, µg/L ^†^	10761.8 ± 10805.7 (*n* = 113)	7899.8 ± 9314.7 (*n* = 27)	11660.3 ± 11130.8 (*n* = 86)	0.12
Highest PCT, ng/L	6.53 ± 16.39	0.55 ± 0.53	8.38 ± 18.39	0.02 *
Highest CRP, mg/L	211.6 ± 122.1	176.5 ± 116.3	222.4 ± 122.4	0.07
Highest WBC count, 10^9^/L	22.0 ± 20.4	15.6 ± 6.6	24.0 ± 22.7	0.04 *
Highest PLT count, 10^9^/L	262.8 ± 168.1	330.5 ± 182.9	242.1 ± 158.6	0.01 *
Highest AST, µkat/L ^2^	6.12 ± 18.76	1.42 ± 0.88	7.57 ± 21.27	0.12
Highest ALT, µkat/L ^2^	3.19 ± 7.85	1.80 ± 0.93	3.63 ± 8.94	0.27
Highest GGT, µkat/L ^3^	3.41 ± 3.55	3.03 ± 2.10	3.53 ± 3.89	0.50
Highest bilirubin, µmol/L	13.72 ± 9.63	10.83 ± 4.49	14.62 ± 10.59	0.06

Values represent means with standard deviations. ^†^ Number in brackets represents the number of patients who had that measurement taken. * Denotes statistically significant difference between groups at <0.05 level. ^1^ Conversion factor (CF): 1 kPa = 7.5 mmHg. ^2^ CF: 1 µkat/L = 60.24 unit/L. ^3^ CF: 1 µkat/L = 59.99 unit/L. Abbreviations: ALT, alanine transaminase; AST, aspartate aminotransferase; BNP, brain natriuretic peptide; CRP, C-reactive protein; DAP, diastolic arterial pressure; FiO_2_, fraction of inspired oxygen; GGT, gamma-glutamyl transferase; ICU, intensive care unit; PaCO_2_, partial arterial carbon dioxide pressure; PaO_2_, partial arterial oxygen pressure; PCT, procalcitonin; PLT, platelets; SAP, systolic arterial pressure; StHbO2, oxygen saturation; WBC, white blood cells.

**Table 3 viruses-17-00244-t003:** Respiratory support, specific treatment modalities, and complications.

Variable	All (*n* = 130)	Long-Term Survivors (*n* = 30)	Long-Term Non-Survivors (*n* = 100)	Statistics (*p*)
Ventilatory support				
Self-proning, *n* (%)	5 (4)	2 (7)	3 (3)	0.33
High-flow, *n* (%)	43 (33)	11 (37)	32 (32)	0.66
Duration of high-flow, days	2.3 ± 1.7	2.1 ± 1.8	2.4 ± 1.7	0.60
NIV, *n* (%)	41 (32)	4 (13)	37 (37)	0.01 *
Duration of NIV, days	2.6 ± 2.2	2.2 ± 1.0	2.6 ± 2.3	0.73
IMV, *n* (%)	88 (68)	16 (53)	72 (72)	0.07
Duration of IMV, days	10.4 ± 10.2	7.1 ± 8.3	11.1 ± 10.5	0.16
Proning during IMV, *n* (%)	18 (14)	4 (13)	14 (14)	1.00
Maximal PEEP, cmH_2_O	11.5 ± 4.1	11.9 ± 3.4	11.4 ± 4.3	0.73
Tidal volume, mL	512.0 ± 143.8	521.2 ± 121.0	510.1 ± 149.0	0.82
Highest peak pressure, cmH_2_O	33.6 ± 9.9	32.9 ± 5.5	33.7 ± 10.5	0.81
Medical and renal support				
Methylprednisolone, *n* (%)	130 (100)	30 (100)	100 (100)	1.00
Levosimendan, *n* (%)	9 (7)	0 (0)	9 (9)	0.12
Nitric oxide inhalation, *n* (%)	5 (4)	2 (7)	3 (3)	0.33
Thrombolysis (rTPA), *n* (%)	3 (2)	0 (0)	3 (3)	1.00
Renal replacement therapy, *n* (%)	18 (16)	4 (7)	14 (24)	0.01 *
Complication rate				
VAP, *n* (%)	45 (35)	6 (20)	39 (39)	0.08
CRBSI, *n* (%)	1 (1)	0 (0)	1 (1)	1.00
Urosepsis, *n* (%)	24 (18)	3 (10)	21 (21)	0.23
Fungal infection, *n* (%)	35 (27)	4 (13)	31 (31)	0.06

Values represent means with standard deviations or number of subjects with percentages. * Denotes statistically significant difference between groups at <0.05 level. Abbreviations: CRBSI, catheter-related bloodstream infection; IMV, invasive mechanical ventilation; NIV, non-invasive mechanical ventilation; PEEP, positive end-expiratory pressure; rTPA, recombinant tissue plasminogen activator; VAP, ventilator-associated pneumonia.

**Table 4 viruses-17-00244-t004:** Point-of-care heart and lung ultrasound data.

Variable	All (*n* = 130)	Long-Term Survivors (*n* = 30)	Long-Term Non-Survivors (*n* = 100)	Statistics (*p*)
Heart				
LVEF, %	51.4 ± 14.4	56.5 ± 4.9	50.2 ± 15.6	0.11
LVOT VTI	18.7 ± 4.9	20.2 ± 4.5	18.4 ± 4.9	0.16
sPAP, mmHg	37.8 ± 12.7	30.4 ± 9.2	39.1 ± 12.8	0.04 *
TAPSE, mm	19.5 ± 5.0	22.5 ± 3.5	18.7 ± 5.1	<0.01 *
TAPSE/sPAP, mm/mmHg	0.52 ± 0.22	0.77 ± 0.21	0.47 ± 0.19	<0.01 *
VCI min. diameter, cm	1.6 ± 0.8	1.4 ± 0.7	1.6 ± 0.8	0.59
VCI max. diameter, cm	2.1 ± 0.5	2.0 ± 0.2	2.1 ± 0.6	0.86
Lung				
Diffuse B-lines pattern, *n* (%)	62 (50)	14 (52)	48 (48)	1.00
Mixed A-/B-lines pattern, *n* (%)	44 (34)	11 (40)	33 (33)	0.83
A-lines pattern, *n* (%)	3 (2)	0 (0)	3 (3)	1.00
Pleural effusion, *n* (%)	18 (14)	2 (7)	16 (16)	0.24
Lung consolidations, *n* (%)	4 (3)	1 (3)	3 (3)	1.00

Values represent means with standard deviations or number of subjects with percentages. * Denotes statistically significant difference between groups at <0.05 level. Abbreviations: LVEF, left ventricular ejection fraction; LVOT VTI, left ventricular outflow tract velocity time integral; TAPSE, tricuspid annular plane systolic excursion; VCI, vena cava inferior.

**Table 5 viruses-17-00244-t005:** Cox proportional-hazards regression survival analysis.

Variable in the Model	HR	95% CI	Statistics (*p*)
Univariate model of long-term mortality (at ICU admission)			
sPAP ≥ 35 mmHg	1.74	1.02–2.99	0.043 *
Model with age as a cofactor (Overall model Chi-squared *p* = 0.040)			
sPAP ≥ 35 mmHg	1.77	1.03–3.04	0.038 *
Age	1.05	0.99–1.12	0.128

* Denotes statistical significance at <0.05 level. Abbreviations: CI, confidence interval; ICU, intensive care unit; HR, hazard ratio; sPAP, systolic pulmonary artery pressure.

**Table 6 viruses-17-00244-t006:** ROC curve analysis of PAPS as long-term mortality predictor.

	AUC	95% CI	Statistics (*p*)
Long-term mortality	0.697	0.571–0.805	0.0265 *

* Denotes statistical significance at < 0.05 level. Abbreviations: AUC, area under the curve; CI, Confidence interval; ROC, receiver operating characteristic.

## Data Availability

The datasets used and/or analyzed during the current study are available from the corresponding author on reasonable request.

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
