# Peer review of "Systolic Pulmonary Artery Pressure as Long-Term Mortality Predictor in Elderly Critically Ill with Severe COVID-19 Pneumonia"

_viruses, 2025, doi:10.3390/v17020244_

Round 1

Reviewer 1 Report

Comments and Suggestions for Authors

This manuscript explores the association between echocardiographically measured systolic pulmonary artery pressure (sPAP) at ICU admission and long-term mortality in elderly patients with severe COVID-19 pneumonia, demonstrating potentially important clinical implications for risk stratification and management. Overall, the study is of interest; however, there are several points that would benefit from further clarification and elaboration:

1.       The reported PASP values are not particularly high. With an echocardiographic sPAP around 40 mmHg, the corresponding mean pulmonary artery pressure (mPAP) assessed by right heart catheterization could very well be less than 20 mmHg, which does not strictly meet the usual cut-off for pulmonary hypertension (PH).

2.       While echocardiography is a useful tool, it carries inherent measurement limitations. It would be helpful to know whether any of the patients underwent invasive hemodynamic monitoring so that sPAP values could be validated against catheter-based measurements.

3.       The potential mechanisms underlying elevated pulmonary artery pressures are not explored in sufficient detail. Distinguishing among hypoxia, pulmonary embolism, or left heart dysfunction as contributing factors would strengthen the discussion of causality.

4.       The study would benefit from more comprehensive echocardiographic data, particularly regarding right ventricular and right atrial dimensions, which could provide greater insight into right heart remodeling and dysfunction.

5.       For the 30 long-term survivors, it would be informative to report any follow-up echocardiographic assessments to evaluate whether pulmonary artery pressures normalized or remained elevated over time.

Reviewer 2 Report

Comments and Suggestions for Authors

The Authors have described in a retrospective study a series of cases of patients with COVID-19 and admitted to intensive care evaluated with echocardiography to identify the presence or absence of pulmonary hypertension. The title of the study "Systolic pulmonary arterial pressure as a predictor of long-term mortality in critically ill elderly with severe COVID-19 pneumonia" photographs exactly what the Authors have described in their text. The age of the patients considered is over 70 years and for this reason the study is even more interesting. The high didimer considered is not statistically significant in the acute phase of COVID-19 in this series of cases. The authors' work is accurate and concludes for a greater short- and long-term survival for patients without pulmonary hypertension.

The text is written in good English and is understandable.

The data reported in this study can help to shed more light on the context of the acute phase of COVID-19 in relation to pulmonary hypertension in elderly patients. In conclusion, the article presents some interesting elements regarding COVID-19 mortality associated with increased pulmonary hypertension for clinical management and further research in this area.
